# Adhesion of Bis-Salphen-Based Coordination Polymers to Graphene: Insights from Free Energy Perturbation Study

**DOI:** 10.3390/polym14214525

**Published:** 2022-10-26

**Authors:** Sergey Pyrlin, Veniero Lenzi, Alexandre Silva, Marta Ramos, Luís Marques

**Affiliations:** Physics Center of Minho and Porto Universities (CF-UM-UP), University of Minho, Campus de Gualtar, 4710-057 Braga, Portugal

**Keywords:** salphen base, metal-organic, molecular dynamics, free energy, self-assembly

## Abstract

Manipulation of nanoscale objects using molecular self-assembly is a potent tool to achieve large scale nanopatterning with small effort. Coordination polymers of bis-salphen compounds based on zinc have demonstrated their ability to align carbon nanotubes into micro-scale networks with an unusual “rings-and-rods” pattern. This paper investigates how the compounds interact with pristine and functionalized graphene using density functional theory calculations and molecular dynamic simulations. Using the free energy perturbation method we will show how the addition of phenyl side groups to the core compound and functionalization of graphene affect the stability, mobility and conformation adopted by a dimer of bis-(Zn)salphen compound adsorbed on graphene surface and what it can reveal about the arrangement of chains of bis-(Zn)salphen polymer around carbon nanotubes during the self-assembly of microscale networks.

## 1. Introduction

The last decades brought significant breakthroughs in nanoscale synthesis and property engineering [1,2,3]. However, exploiting these advances often requires positioning, alignment of nanoscale objects and assembling them into higher order structures, such as lattices of nanocrystals for thermoelectrics [4,5], networks of nanofillers for optoelectronics and photovoltaic applications [6,7] etc.

One particularly interesting way of constructing interconnected networks of nanotubes was reported by Escarcega-Bobadilla et al. [8], who observed that certain metal organic compounds can self-assemble into networks upon drop-casting from dichloromethane (DCM) upon solvent evaporation (Figure 1a). The networks were composed of microscale rings and rods interconnected in a neuron-like fashion. While the thickness of the observed rings and rods was estimated to be around 100 nm, the networks remained connected across areas of hundreds of micrometers. Even more intriguing, such self-assembling networks have demonstrated an ability to incorporate carbon nanotubes (CNTs) and align them both on surface and in the bulk of polymer, which resulted in significant boost of electrical conductivity. While both the arrangement of carbon nanotubes into circular patterns [9,10] and self-assembly of polymers and metal-organic compounds into circular structures [11,12] has been studied previously, the self-assembly of bis-(Zn)salphen compounds deserves a special attention both from fundamental point of view due to its unusual neuron-like pattern of interconnected “rings-and-rods” as well as from application standpoint as an approach to produce conductive networks from very diluted solutions of conductive fillers.

The chemical structure of the compounds, for which self-assembly of networks was achieved, is shown in the Figure 1b: it consists of the two symmetrical salphen bases binding metal (Zn) cations through coordination interaction with oxygen and nitrogen atoms, connected with central biphenyl bond and functionalized with additional phenyl rings. 

A subsequent study [13] has explained this phenomenon by the formation of a coordination polymer, the periodic units of which are dimers of the bis-(Zn)salphen compounds (Figure 1c). Using atomistic simulations it was shown that such chains can form semiflexible fibers via π-π interactions due to phenyl side groups. It is expected that upon solvent evaporation such fibers aggregate and form networks of interconnected loops and toroidal globules in accordance with experimental observations.

However, the interaction of bis-(Zn)salphen compounds and carbon nanotubes, more specifically their incorporation in the core of the self-assembled networks, remains unexplored. This aspect is especially crucial for applications as electrical conductivity of CNT-based films and composites depends drastically on the contact between nanotubes [14,15]. Furthermore, molecular rearrangement in the nanoscale gaps between carbon nanotubes can alter mechanical and transport properties of surrounding matrix material [16]. In this paper we will provide an insight to this phenomenon from atomistic calculations.

The free energy is a key quantity to understand the physical processes in a solvent environment, such as conformational changes, protein-ligand binding and molecule-substrate adsorption [17]. A numerical estimate of the free energy difference ($\Delta G_{i\to f}$) between two states *i* and *f*, characterized by potential energies $U_{i}\;$and $U_{f}$, close enough in the configuration space, is given by the Zwanzig formula [18], which is at the basis of many free energy estimation methods:(1)$$\Delta G_{i\to f}=-k_{B}T\;ln\langle exp(-\frac{\left(U_{i}-U_{f}\right)}{k_{B}T})\rangle ,$$
where *T* is the ambient temperature, $k_{B}$ is the Boltzmann constant and <> denotes an ensemble average by the microstates of the starting state *i*. The main factor hindering the application of Equation (1) is high computational cost required for its calculation, as a good sampling over the configuration space is required. As the Zwanzig equation is valid only for states that are close enough in the configuration space, a typical transition needs to be split into many intermediate steps for which Equation (1) holds, and in doing so the free energy along the resulting path can be calculated, which is referred to as the potential of mean force (PMF). For instance, Umbrella Sampling method [19,20] consists of simulations of the system dynamics, constrained by an external elastic force to one of a set of overlapping windows, subdividing the path from initial to final state. Then a weighted histogram analysis method (WHAM) [21] is used to reconstruct PMF, taking into account the restraining potentials applied within each window. Another way to estimate $\Delta G_{i\to f}\;$is to estimate the work needed to force the system from one state to another ($W_{i\to f}$) in an non-equilibrium process, which is related to a free energy change via Jarzynski’s equality ($exp\left(-\Delta G_{i\to f}/k_{B}T\right)$= $\langle exp(-W_{i\to f}/k_{B}T)\rangle$) [22]. Then one could perform a simulation including a forcing term to drive the dynamics and track the forces and displacements to estimate the work. This approach is called steered molecular dynamics (SMD) [23,24]. On the other hand, the adaptive biasing force method [25] implies altering the free energy landscape of a process in such a way to nullify any difference in free energy along the path. This way, the PMF is reconstructed from the amount of biasing potential used.

Alternatively, the free energy perturbation method (FEP) [26,27,28] suggests to extend the systems potential energy into a function of a coupling parameter (λ) in such a way that it gradually changes between the initial and final states: $U\left(\lambda \right)=\lambda U_{f}+\left(1-\lambda \right)U_{i}$. In this case the transition consists of simulations of system dynamics under potential corresponding to a set of intermediate values of λ, rather than forcefully drugging the molecules along a predefined path. The transition free energy can then be evaluated using the Bennett acceptance ratio method [29,30]. FEP was shown to give very precise results with the relative errors of the order of 1–2 kcal/mol [26].

An intriguing way to use FEP to reduce the computational cost of free energy calculations bases on the fact that the Gibbs free energy differences depend only on the initial and final states and not on the path connecting them. Such approach, known as alchemical transformations [31] substitutes estimating $\Delta G_{i\to f}$ of a transition that can be harder to trace directly by introducing an intermediate state, related to the initial and final states by transitions, which may be unphysical, but computationally more convenient. Thus, solvation of a new compound can be studied by placing both solute and solvent molecules into the same simulation box and vary the strength of the solute-solvent interaction potential by using a coupling parameter λ as a scaling factor. Scaling interaction up (λ=0→1) will correspond to immersion of a solute into solvent (solvation). In certain cases, however, it is more convenient to model the opposite process—decoupling (or extraction) of solute from solvent (λ=1→0). Thus, free energy of a reaction in solvent can be estimated using the solvation or decoupling free energies of the reactants and products and free energy of the same reaction in vacuum. This approach has been successfully used in computational drug discovery simulations to determine drug-ligand binding free energies [32,33], as well as to investigate the exfoliation properties of 2D materials in various solvents [34].

In this study we combine density functional theory and free energy perturbation calculations to compare possible mechanisms of adhesion of Zn-salphen compounds to carbon nanotube surface. Given a significant difference in scale between CNT diameter (tens of nm) and size of a bis-(Zn)salphen compound (>2 nm), we replace nanotube surface with a periodic sheet of plane graphene. Focusing on the atomistic details of coordination polymer adhesion to CNT we model the interacting tip of a polymer chain with its single building block—a dimer of bis-(Zn)salphen compounds. In the following sections we will describe the calculation setup used in our simulations and the effect of compound and graphene functionalization on the strength of adhesion and geometry of the adsorbed compounds in dichloromethane solvent.

## 2. Materials and Methods

### 2.1. Ab Initio Energy of Adhesion

To estimate the energy of adhesion of the (Zn)salphen-based compounds to graphene surface we have performed *ab initio* calculations for the systems consisting of single salphen base compound of Zn and a periodic sheet of pristine or functionalized graphene. Since the chemical structure of a salphen ligand is rich in aromatic rings, π-π interaction is a primary factor of adhesion. However, compound’s cation center surrounded by negative oxygen and nitrogen atoms would be attracted to charged impurities on graphene surface. For this reason, besides pristine graphene we include sheets modified with: boron and nitrogen substitutions, chemisorbed oxygen, hydroxyl and carboxyl groups, which have been reported to improve graphene adhesive properties [35,36,37].

System under study (Figure 2) was enclosed in (approximately) 2.48 × 2.58 × 1.29 nm periodic box, which is enough to accommodate a supercell of graphene with 240 atoms (6 by 10 rectangular unit cells of the hexagonal graphene structure) and ensure lack of interaction of the compound with its periodic images.

The calculations were performed with the SIESTA DFT package (v. 4.1) [38]. Exchange and correlation functionals were described by generalized gradient approximation in Perdew-Becke-Ernzerhof approximation [39]. Valence electrons orbitals were simulated by a basis set of triple-zeta triply polarized numerical atomic orbitals with energy shift 10 meV [40]. Core electrons were replaced by Troullier-Martins pseudopotentials [41]. Cut-off of 1000 Ry was used to define integration mesh. Van der Waals interaction was included using Grimme potentials [42]. Counterpoise correction to exclude basis superposition error [43]. Energy minimization was performed with respect to atomic positions and lengths of periodic box sides. Geometry of standalone graphene and salphen-Zn compound was optimized until total energy was converged to 0.001 eV, maximum atomic force did not exceed 0.01 eV/Å and maximum atomic displacement was 0.01 Bohr. Correspondingly, the weakly interacting graphene–salphen system was converged to 0.01 eV, 0.1 eV/Å and 0.05 Bohr.

The set of calculations included geometry optimizations to obtain relaxed structures followed by single point energy calculations to get ground state energies of the standalone compound ($E_{g}^{C}$) and graphene sheet ($E_{g}^{G}$), as well as of the complex of the compound adsorbed on the graphene surface ($E_{g}^{C+G}$). Furthermore, additional single point energy calculations were performed for each of compound and graphene in the geometry of the complex to estimate the basis superposition error for each counterpart ($E_{BSSE}^{C,G}$). Then the energy of adhesion for graphene-compound system can be calculated according to:(2)$$E_{adh}=\left(E_{g}^{C+G}-E_{g}^{C}-E_{g}^{G}\right)-\left(E_{BSSE}^{C}+E_{BSSE}^{G}\right).$$

### 2.2. Molecular Dynamic Simulations

To estimate the free energy of adhesion of bis-(Zn)salphen dimers to graphene surface we have performed all-atom free energy perturbation simulations with LAMMPS package [44]. The solvent (Dichloromethane, DCM) molecules were treated explicitly. Non-bonded (van der Waals and electrostatic) interactions were modelled using 6–12 Lennard-Jones and Coulomb potentials. A cut-off of 10 Å was used for short range part of both potentials. The long range interaction was treated using particle-particle particle-mesh solver [45]. Electrostatic interactions were modelled using fixed atomic charges, fitted by RESP method [46]. Electrostatic potential grid was obtained from ab initio calculations using Gaussian package [47]. The interactions between Zn^2+^ ion and organic ligand were treated within a free cation model [48] in order to allow the compound to adopt more relaxed conformation while interacting with hydroxyl and carboxyl groups of functionalized graphene [46]. During FEP simulation “soft” version of van der Waals potential was used to gradually scale down the strength of interactions without creating a large deviation from equilibrium [49].

Both covalent and van der Waals interactions were described by OPLS-AA force field [50,51], with the exception for the atoms of the imine group (-C=N-), explicit parametrization of which was introduced in OPLS_2005 force field [52] and is essential for accurate description of salphen compounds. While many improvements were introduced over the years with more recent parametrizations [53,54,55,56], such as more precise torsions for DNA backbone chain and better description of nucleic acid interaction, the OPLS-AA force field is still widely used for molecular simulations in liquid state, such as including recent studies of SARS-CoV-2 inhibitors [57], chromophores [58] and electrolytes for novel batteries [59]. It is also implemented in many open source toolkits for building input files for molecular dynamics simulations, such as Moltemplate [60], used in this work. Although virtual charge sites, introduced in latest parametrizations [55,56] could further improve the description of hydrogen bonding to functional groups—this will be investigated in future work.

The system under study consisted of a dimer of bis-(Zn)salphen compound, put at random orientation on top of 4 layered graphene sheet of 12 by 20 rectangular unit cells of the hexagonal graphene structure (approx. 5 × 5 nm), periodic in 2 in-plane directions (*x* & *y*). Due to the symmetry of the bis-salphen dimer the number of its orientations relative to graphene surface which needs to be studied can be reduced. To obtain initial positions the dimer was placed in one of the orientations shown in the Figure 3 and rotated randomly around the axis perpendicular to the graphene surface (*z*). The dimer was placed in each orientation and relaxed 5 times, resulting in 20 independent simulations.

For each independent run the movement of atoms of the dimer were first simulated in vacuum at 300K then laid down onto graphene surface and relaxed. After which the rest of the space on top of graphene was filled with DCM molecules up to 5 nm in height. The initial positions of solvent molecules were taken from an independent simulation of dichloromethane squeezed between graphene in the same simulation box to ensure close to equilibrium initial conditions, the solvent molecules intersecting with the compound molecules were excluded. Temperature and pressure of solvent were controlled by Nose-Hoover thermostat and barostat [61], with the atmospheric pressure applied in the *z* direction, perpendicular to the graphene surface, while the simulation box bounds the two in-plane directions were kept fixed. The motion of the atoms of dimers and graphene were controlled separately by Langevin thermostats [62].

Furthermore, during the first 500 fs of solvent equilibration elastic restraints with gradually decreasing spring constants were applied to the heavy atoms of the compound to avoid the initial relaxed state being compromised by excessive stress from unequilibrated solvent. After initial stress relaxation and equilibration during 250 ps with timestep of 0.5 fs, the movement of hydrogen atoms was simulated using SHAKE algorithm [62] and the timestep increased to 2 fs for a 2 ns equilibration run. Besides that, to get the reference solvated state the same bis-salphen dimer was placed in a 5 × 5 × 5 nm solvent box without graphene for another 5 independent simulations. Periodic boundary conditions were applied to all surfaces of simulation box.

After initial 2 ns equilibration a FEP simulation was conducted for each system. The simulation consisted of the two stages. On the first the electrostatic interaction between the compound and surrounding environment was gradually scaled down during approximately 2.0 ns of simulation time (20 intermediate λ-values each simulated during 0.1 ns). On the second stage van der Waals interactions between the atoms of compound and environment were scaled down. To make the transition smoother on the second stage the compound was decoupled atom by atom starting from exterior hydrogen atoms and finishing with the core Zn, O, N atoms. For each atom 10 intermediate λ-values were used, each simulated during 0.025 ns. Total simulation time was 40–50 ns depending on the variation of the bis-salphen compound (with or without phenyl groups). Every 10 simulation steps (20 fs) backward and forward difference of interaction potential was recorded and subsequently processed using the Bennett acceptance ratio method to estimate accumulated change of Gibbs free energy ΔG. Accumulated change of the Gibbs free energy was estimated from the recorded backward and forwards differences of interaction potential using ParseFEP plugin of VMD [27,63].

The thermodynamic cycle corresponding to this study is shown in the Figure 4. The red arrow indicates the transition of interest between solvated (c) and adsorbed compound (a) as the free energy change between these states contains the free energy of adhesion ($\Delta G_{adh}$). While it is possible to estimate it directly by steered molecular dynamics, due to large capacity of the compound dimer to deform and change conformation a large number of slow simulations would be required to accurately sample the PMF change during slow approximation or detachment of the compound molecules. Instead, we estimate the $\Delta G_{adh}$ indirectly by:(3)$$\Delta G_{adh}+\Delta G_{G}^{solv}=-\left(\Delta G_{S\&G}^{dec}+\Delta G_{G}^{dec}+\Delta G_{S}^{solv}\right),$$
where $\Delta G_{G}^{solv}$—free energy of solvation of graphene sheets, $\Delta G_{S\&G}^{dec}$—free energy change during decoupling of the adsorbed compound dimer from surrounding solvent and graphene molecules, $\Delta G_{G}^{dec}$—free energy change during decoupling of the graphene from solvent and $\Delta G_{S}^{solv}$—free energy of solvation of the compound. Since within FEP formalism, decoupling of the compound from solvent is the opposite of solvation (immersion into solvent), the solvation free energy is the negative of the decoupling free energy: $\Delta G_{}^{solv}=-\Delta G_{}^{dec}$. Using this substitution in Equation (3) the desired free energy of adhesion to graphene in solvent can be estimated as follows:(4)$$\Delta G_{adh}^{}=\Delta G_{S}^{dec}-\Delta G_{S\&G}^{dec}.$$

Using the decoupling transition to estimate the free energy of adhesion in this case is technically simpler than solvation (immersion) as in the latter process it is not guaranteed that during finite simulation time the compound will adopt the desired conformation on top of the graphene surface. On the contrary, in the case of decoupling process both starting and final states are well defined and if the set of initial positions of the adsorbed compound molecules contained the most favorable position, the largest estimated $\Delta G_{}^{dec}$ across the simulations will produce a good estimate of $-\Delta G_{adh}^{}$.

A slightly different approach was used in case of the functionalized graphene (Figure 5). Rather than simulating again the lengthy decoupling of the compound dimer from graphene and solvent, we have estimated the free energy change during removing the functional groups (-OH and -COOH) from graphene and adsorbed compound ($\Delta G_{F*}^{dec}$). In this case both Coulomb and van der Waals interactions were scaled (up or down) simultaneously for all group atoms through 20 intermediate λ-values, simulated during 0.1 ns each.

In this case, approximating the free energy of immersion of the compound dimer on top of functionalized graphene by $\Delta G_{S\&G}^{solv}+\Delta \Delta G_{adh}$, where $\Delta \Delta G_{adh}$ is the correction to adhesive energy due to interaction of compound with the functional group, the transition free energies are linked with the following relation:(5)$$\Delta G_{S\&G}^{solv}+\Delta \Delta G_{adh}=-\left(\Delta G_{F*}^{dec}+\Delta G_{S\&G}^{dec}+\Delta G_{F}^{solv}\right).$$Using the same equality between solvation and decoupling free energies, it is straightforward from (5) that the adhesive energy correction can be estimated as:(6)$$\Delta \Delta G_{adh}=-\left(\Delta G_{F*}^{dec}+\Delta G_{F}^{solv}\right).$$
where $\Delta G_{F}^{solv}$—is the free energy change due to immersion of the functional group on top of graphene in solvent. Since the position of the functional group is well defined by a chemical bond to graphene, here we calculate $\Delta G_{F}^{solv}$ directly, rather than through decoupling process as we did for compound solvation. Furthermore, immersion (in case of $\Delta G_{F}^{solv}$) and decoupling (in case of $\Delta G_{F*}^{dec}$) of both electrostatic and van der Waals interactions are performed simultaneously during 2 ns simulations. A set of 10 immersion and 20 decoupling simulations were performed for each of the selected functional groups.

The initial guesses for the positions of the bis-salphen dimer, interacting with the functional groups were obtained by placing of the separately optimized geometries of the dimers so that the coordinates of the Zn, O and N atoms of one of its non-bonded salphen groups closely approximate the position of the corresponding atoms of the single Zn-salphen base interacting with -OH or -COOH functional group of graphene patch from DFT study. To obtain the relaxed initial positions vacuum simulations were performed in which the hydrogen and oxygen atoms of the functional groups were initially attached to the oxygen and zinc atoms of the compound with gradually relaxing spring force.

## 3. Results

### 3.1. Binding of a Single Salphen Complex to Functionalized Graphene

To compare the influence of functionalization on the adhesion of the (Zn)salphen-based compounds to graphene surface we have performed *ab initio* calculations for a single compound adsorbed on the surface of pristine and functionalized graphene as shown in the Figure 2. These calculations served to analyze which type of interactions is mostly responsible for the adhesion and select the systems of interest for molecular dynamics study. The optimized geometries of the compound-graphene system are shown in the Figure 6. The estimated energies of adhesion are summarized in the Table 1.

On the surface of a pristine graphene the adsorbed (Zn)salphen compound assumes completely flat shape lying parallel to the graphene surface at the distance of 3.04 Å. This together with the high negative value of the energy of adhesion of a (Zn)salphen compound to pristine graphene, indicates that the π-π interactions between graphene surface and aromatic rings of the salphen base play the major part in attraction of the compound.

Doping graphene with B or N impurities provides a moderate improvement of adhesion due to electrostatic attraction between the impurity and the Zn or O atoms of salphen compound without affecting the π-π interactions: the flat shape of the compound and separating distance of 3.04 Å is maintained.

At the same time, the presence of a carbonyl oxygen, i.e., connected to the two graphene atoms, has an opposite effect. Despite the ability of Zn to form additional coordination bonds, connecting to out-of-plane O^2−^ center of the oxidized graphene requires the salphen compound to leave its favorable flat conformation and overcome the potential barrier created by N and O atoms, surrounding Zn in salphen complex. Ass the result, the average distance between salphen aromatic rings and graphene surface increases to 3.23 Å and adhesion energy decreases by 3.3 kcal/mol.

On the contrary, functionalizing graphene with hydroxyl (-OH) and carboxyl (-COOH) groups makes the compound to bind better to the graphene surface, as these groups form hydrogen bonds with the salphen compound. Hydrogen bond lengths are estimated as 1.75 Å in case of hydroxyl group and 1.50 Å in case of carboxyl group. However, while both groups provide a single hydrogen bond, the benefit is significantly higher in case of carboxyl group. The reason for this is that hydroxyl group is situated closer to the graphene surface and attraction of one of salphen’s oxygens to hydroxyl hydrogen simultaneously with repulsion of the other salphen’s oxygen from the oxygen of the functional group creates a distortion of the plane shape of salphen compound, counteracting with the π-π interactions. On the contrary, the higher position of the carboxyl group and flexibility of rotation of its bonds allows the hydrogen bond to be formed without distorting the π-π interactions: salphen compound maintains its flat shape and distance to graphene of 3.06 Å.

Although the observed improvement of binding energy is comparatively small in comparison with the impact of aromatic interactions in vacuum, in a good solvent, such as dichloromethane (DCM), the molecules of the compound will experience a competing attraction by the solvent molecules. In such case these additional interactions with functionalized graphene may give a significant boost in favor of adsorption.

### 3.2. Free Energy of Adhesion of a Bis-Saplhen Dimer

A better criterion for the tendency of a molecule to be adsorbed on a surface is the Gibbs free energy of adsorption ($\Delta G_{adh}^{}$). Besides potential energy of attraction it includes temperature and entropy contributions, which are crucial to describe the behavior of polymers in solution. Here using the FEP method we compare the effect of adding functional groups to the chemical structure of bis-salphen compounds and graphene surface on the free energy of adhesion between them in dichloromethane solvent (DCM), which was shown to produce the desired self-assembled networks [8].

#### 3.2.1. The Effect of Compound Functionalization

First, we study the adhesion of a dimer of the reference bis-(Zn)salphen compound (compound A, Figure 1b,c) to graphene surface. In the previous study [13] it was shown that the two phenyl groups attached at the side of the compound play a key role in linking the chains of coordination polymer into fibers and sheets. As our *ab initio* calculations show, π-π interactions are mainly responsible for a salphen adhesion to graphene. On the other hand, however, the presence of phenyl side groups distorts the otherwise planar shape of the salphen bases. As DFT calculations have shown this can have significant negative effect on the adhesion through π-π interactions. Therefore, it is instructive to see if these groups also promote adhesion to graphene. For this reason, we compare the free energy of adhesion of the reference compound A with a variation of the same chemical structure without these additional phenyl groups (compound B). For this purpose, for each of compounds we have conducted series of FEP calculations as described in section Materials and Methods. The accumulated change of Gibbs free energy during the transitions are shown in the Figure 7. The initial smooth parabolic segment represents scaling down of electrostatic interactions, the subsequent wavy part is comprised of segments representing decoupling Lennard-Jones potential for individual atoms of compounds.

As in case of the adsorbed compound, the free energy varies significantly with the orientation of the compound dimer towards graphene surface. As can be seen from the individual FEP results for each adsorbed compound (grey lines), the discrepancy of $\Delta G_{adh}^{}$ is of the order of 10 kcal/mol. For this reason, to estimate the free energy of adhesion we use the average of 5 most favorable simulations (highest $\Delta G_{S\&G}^{dec}$) as the estimate of the negative solvation free energy in the Equation (4). The estimated values are collected in the Table 2.

Binding of dimers of bis-(Zn)salphen compound to a graphene surface is favorable (negative $\Delta G_{adh}$) for both compounds. However, contrary to expected, phenyl side groups do not result in a stronger adhesion. Despite the fact that decoupling simulations indeed show larger absolute value of ΔG for adsorbed compound A, the absolute values of free energy of solvation show yet increased affinity of the phenyl functionalized compound to the DCM solvent. To understand this behavior, not expected for an aromatic functional group in non-aromatic solvent, we need to take a closer look on how the compounds interact with dichloromethane solvent and graphene substrate. The Figure 8 shows the adsorbed dimers of both compounds in the conformation for which the highest (by absolute value) adsorption free energy was found. It can be clearly seen that for a compound B (Figure 8b), stripped off the functional group, the π-π interaction with graphene substrate is provided by one of the salphen bases of the lower molecule flattened along the surface and some aromatic rings of the upper molecule. Comparing it to the dimer of the compound A (Figure 8a), one can see that, while it also has one salphen base flattened along the graphene surface, its phenyl functional group faces upwards due to steric repulsion from the other aromatic rings of the core of salphen compound. Some of the aromatic rings of the other half of the lower molecule and one of phenyl functional groups of the upper also interact with graphene surface, but they are not aligned parallel to the surface, therefore providing only a small benefit for adhesion.

On the other hand, the presence of the phenyl functional group has its side effect on the compound interaction with dichloromethane as shown in the Figure 9. DCM is a polar solvent and its molecules are attracted by the oppositely charged O, N and Zn atoms in the core of the salphen compound. As the radial pair distribution function shows in the Figure 8b the interaction of oxygen atoms of the compound and dichloromethane’s hydrogens is more pronounced in case of the functionalized compound A. We attribute this to the effect of the phenyl functional group, which experiencing both steric repulsion with the neighboring aromatic rings and collisions with solvent molecules distorts the planar shape of the salphen base and makes the oxygen of the connected phenolic group more open to the solvent, as shown in the Figure 8a. This effect is diminished when the compound is adsorbed onto graphene. Other pair distribution functions between the atoms of compounds and solvent (provided in Appendix A) do not indicate any difference between compounds.

Therefore, on average phenyl functionalization slightly decreases adhesion of compound to graphene surface. It also introduces a barrier to rotation of an adsorbed dimer making it more difficult for compound to adopt a lower energy conformation. This can be seen in the Table 2 from the larger error estimates of solvation and adhesion energies for the compound A.

#### 3.2.2. The Effect of Graphene Functionalization

Next, we compare the effect of adding hydroxyl and carboxyl groups to graphene, which have been chosen because these functional groups increased the adhesion energy between the salphen dimer and graphene sheet, as shown in Table 1. Since the free energy of adhesion of compound A is already known, the effect of functionalization of graphene can be compared in a simpler way: by estimating the free energy change due to presence of the functional group ($\Delta \Delta G_{adh}$) as calculated by the Formula (6). The accumulated free energy changes are shown in the Figure 10.

Figure 10a shows a non-monotonous shape of the accumulated ΔG over the course of simulation in case of hydroxyl group, which is the result of the interplay between electrostatic attraction, scaling in close proximity of the atoms being immersed or decoupled as −*λ*/(1−λ)), dispersive attraction (~ −*λ*/(1−*λ*)^2^), and core repulsion (~ + *λ*/(1−*λ*)^4^) of the soft Lennard-Jones potential between the surrounding molecules and the functional group [64,65,66]. The same processes occur in case of carboxyl group as well (Figure 10b), however, due to much stronger electrostatic attraction to the solvent molecules, the time dependency of the free energy change looks almost monotonic.

The Appendix A shows a comparison of a FEP simulations of immersion of hydroxyl and carboxyl groups with electrostatic and van der Waals interactions scaled separately. The time dependence of the free energy change is different in this case, providing a clear separation between Lennard-Jones (λ < 0.5) and Coulomb scaling regions (λ ≥ 0.5). However, the net free energy change remains the same.

Table 3 summarizes the results of these calculations. As expected, hydrogen bonds between functional groups and oxygen atoms of salphen bases additionally stabilize the bis-(Zn)salphen compounds on graphene surface. Furthermore, besides a hydrogen bond the oxygen atom of carboxyl group also interacts with zinc cation improving binding to carboxyl group.

The resulting estimate of $\Delta \Delta G_{adh}^{}$ for the effect of -COOH functionalization of 8 kcal/mol is comparable to the adhesion free energy of 9.7 kcal/mol of the compound A to the pristine graphene surface. Since the two values were calculated using different approaches it is hard to estimate the uncertainty between them. Apart from shorter simulation time, another reason for changing the approach, is that the former approach (decoupling the whole bis-salphen dimer from the whole surrounding) is less stable in case of functional group. While adhesion of dimer to a pristine graphene is through van der Waals potential, the adhesion to -OH and -COOH is mostly due to electrostatic interaction, which is turned off first. As the result, without introduction of virtual constraints the dimer is likely to break-off the functional group at a random moment in the beginning of a FEP simulation causing large discrepancy in FEP results between independent runs. Simultaneous decoupling of both van der Waals and Coulomb interactions with the functional group while keeping the dimer attracted to the graphene surface provides increased stability. One way to compare the precision of both approaches would be to compare their results to a well converged PLUMED free energy scan [67], which is beyond the scope of the present work. It is, however, still possible to conclude from the relatively large value of $\Delta \Delta G_{adh}^{}$ that functionalization of graphene with hydroxyl and especially carboxyl groups is comparable to the difference of the adhesion free energy due to initial orientation of the dimer.

The effect of functionalization on the dimer orientation can be also seen in the distributions of the angle (θ) between the dimer long axis and graphene plane (Figure 11). The orientation angle was sampled during the initial equilibration of the 5 selected simulations (showing the most favorable adhesion free energy). On the surface of pristine graphene dimers of both compounds A and B adopt orientation parallel to the surface, such as shown in the Figure 3d,e, characterized by θ values around 0. However, functionalization of graphene causes dimers to adopt more elevated orientation with wide angle distribution for -OH group and peak at ~30 degree for -COOH group. This points to potential change of the arrangement of the bis-(Zn)salphen chains from sticking closely along the graphene surface to be aligned out towards solvent (Figure 3c).

Figure 12 shows the configurations with maximum (by absolute value) adhesion energy of the dimers of compound A adsorbed to functionalized graphene. Both functional groups form hydrogen bonds with oxygen atoms of compound: *r_OH_ =* 1.9 ± 0.2 Å and 1.8 ± 0.1 Å in case of hydroxyl and carboxyl groups correspondingly. In the latter case (-COOH group), carbonyl oxygen (C=O) provides additional attraction for zinc cation of the compound: *r_ZnO_ =* 2.0 ± 0.1 Å. As the FEP results in the Table 3 show, these interactions improve binding of the dimers. However, in order to do so, the functional group must be close to oxygen-metal center of the salphen base, therefore limiting the space for the biphenyl ring of the adjacent salphen base. Therefore “elevated” configurations of dimers, in which biphenyl angle of the lower bis-salphen molecule is close to 90 degrees, become preferrable.

## 4. Discussion

In the earlier publication [13] the two main mechanisms for the association of the bis-(Zn)salphen dimers were compared: through coordination Zn-O interaction and through π-π interactions. Using the approach similar to the one used in this work the free energy of dimer binding in DCM was estimated to be ~18 ± 4 kcal/mol in case of the former mechanism and ~5 kcal/mol in case of the latter. Although these estimates were obtained using a different force field (GAFF) and λ-schedule was used during the FEP simulations in this work, it is worth noting that the free energy of adhesion of dimeric units to graphene surface estimated here is close in magnitude to the former result. This lets us to make a conclusion that while bis-(Zn)salphen compounds self-assemble in solution into diluted chains of coordinative polymer, they can also stick to the surface of the nanotubes.

The consequence of this is twofold. Firstly, CNTs or graphene sheets can serve as centers of condensation for bis-salphen chains, which first stick to their large surface, then slide along their surface due to low friction in π-π interacting systems until entangling with other adsorbed chains.

Secondly, long chains, partially adsorbed to the surface of CNTs not only stabilize the nanotubes in solution [68], but also exert random forces due to thermal fluctuations, pulling nanotubes together (“entropic pulling”) [69]. This momentum transfer from condensing chains to diluted CNTs should be especially strong in case of hydroxyl and carboxyl groups, frequently encountered on the surface of oxidized or acid treated CNTs [70,71,72]. Such groups can serve as anchors due to hydrogen bonds and zinc-oxygen interaction.

On the latter stage, a stronger binding of bis-salphen compounds together and mutual attraction of nanotubes’ extended conjugated surfaces, promotes phase separation of the condensed mixture of CNTs and coordinated polymer chains, producing self-assembled network with the core of well aligned nanotubes covered with a shell of aggregated metal-organic chains, as appears on the microphotographs of the original paper [8].

These conclusions lay the ground for any subsequent study of self-assembly of nanotube—coordination polymer networks, as well as their mechanical and transport properties, which must consider orientation and alignment of metal-organic compounds around CNTs and the role of hydrogen bonds.

## Figures and Tables

**Figure 1 polymers-14-04525-f001:**
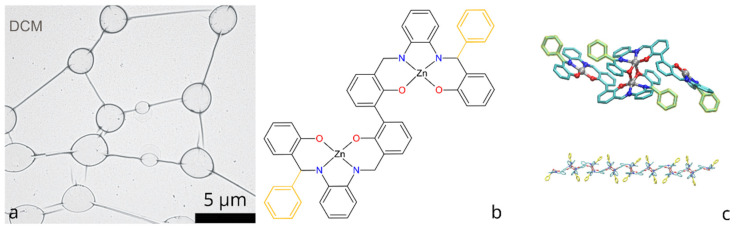
(**a**) Self-assembled molecular network of bis-(Zn)salphen compound, obtained by drop-casting from DCM (preprinted with a permission from Ref. [8]); (**b**) Chemical structure of bis-(Zn)salphen compound, functional phenyl rings are highlighted with yellow; (**c**) 3D rendering of a coordination bonded dimer of the compound and a fragment of a polymer chain of such dimers Ref [13].

**Figure 2 polymers-14-04525-f002:**
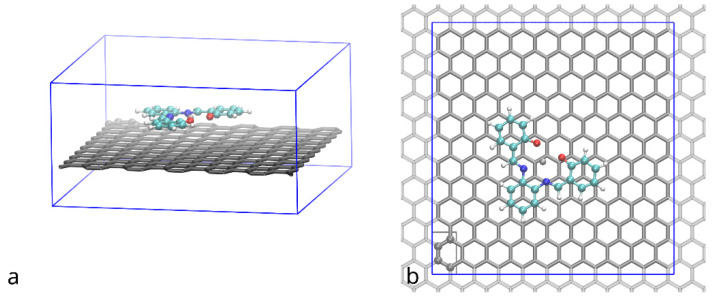
Supercell of graphene (grey tubes) with adsorbed (Zn)salphen compound (ball&stick, color by element) used in DFT simulations: (**a**) side view, (**b**) top view. Periodic image of the graphene beyond the simulation region (blue box) lattice is shown with light grey. Graphene atoms shown with ball and stick representation enclosed in a grey rectangle indicate a rectangle unit cell of a graphene lattice.

**Figure 3 polymers-14-04525-f003:**
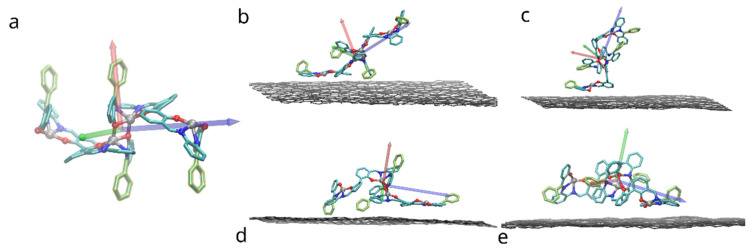
Placement of the bis-(Zn)salphen dimer with functional phenyl rings (highlighted with yellow) atop of graphene surface: (**a**) a compound dimer with axes indicating orientation, (**b**–**e**) variants of placement (after relaxation), axes indicate rotation of dimer. Only topmost graphene layer is shown for clarity, also solvent molecules and hydrogen atoms of compound are omitted.

**Figure 4 polymers-14-04525-f004:**
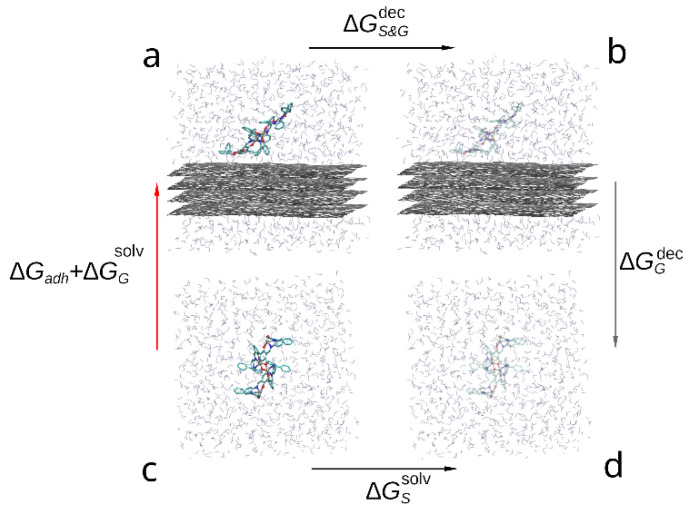
Thermodynamic cycle used to compare the adhesion free energy of the compound to graphene in solvent: (a) a bis-(Zn)salphen dimer interacting with graphene surface, (b) graphene with the dimer decoupled (shown with transparency), (c) dimer solvated in DCM, (d) dimer decoupled from solvent. The red arrow (c-a) indicates the transformation, the free energy of which contains the target contribution, black arrows (a-b and d-c) indicate transformations which are simulated with FEP method and the gray arrow (b-d)—transformation, the free energy of which is cancelled out.

**Figure 5 polymers-14-04525-f005:**
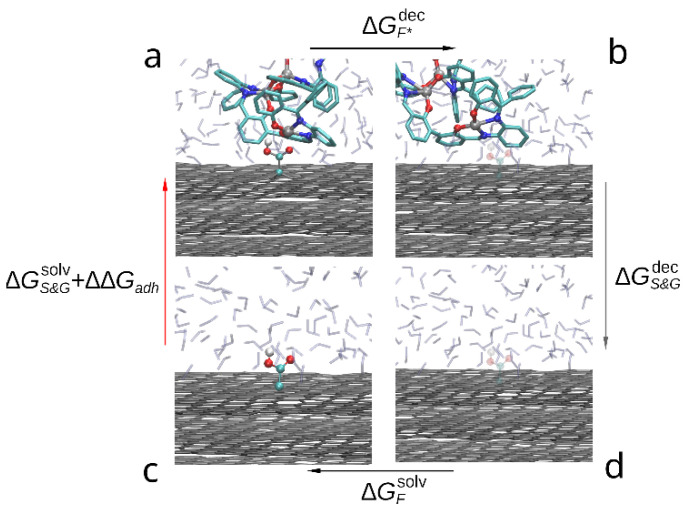
Thermodynamic cycle used to compare the adhesion free energy of the compound to functionalized graphene in solvent: (a) a bis-(Zn)salphen dimer interacting with functionalized (-COOH) graphene, (b) dimer and graphene with the functional group decoupled (shown with transparency), (c) functionalized graphene surface in DCM solvent, (d) graphene surface with the functional group decoupled from surrounding. The red arrow (c-a) indicates the transformation, the free energy of which is the target, black arrows (a-b and d-c) indicate transformations which are simulated with FEP method and the gray arrow (b-d)—transformation, the free energy of which is cancelled out.

**Figure 6 polymers-14-04525-f006:**
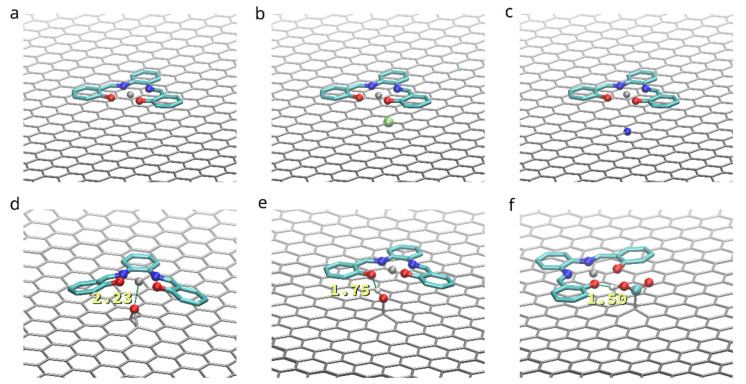
Found relaxed geometries for a single base Zn-salphen compound adsorbed to a pristine graphene (**a**), as well as boron (**b**) and nitrogen (**c**) substituted, oxidized (**d**) and functionalized with hydroxyl (**e**) and carboxyl (**f**) groups. Annotations in (**e**) and (**f**) show hydrogen bond lengths.

**Figure 7 polymers-14-04525-f007:**
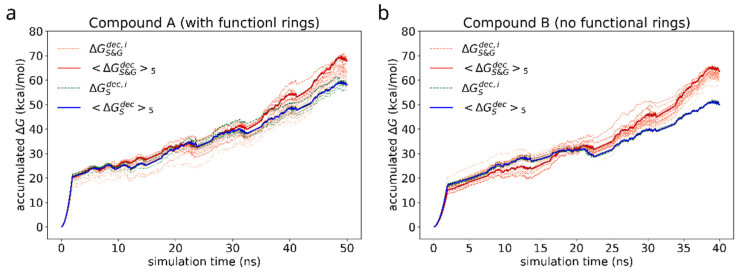
Accumulated free energy change during decoupling of the solvated (green dashed lines) and adsorbed (orange dotted lines) compound dimers. The intensity of the color highlights how close the free energy change of a particular simulation is to the maximum value over the whole set. Blue and red solid lines represent the average of N = 5 most favorable simulations for the cases of solvated and adsorbed dimers correspondingly.

**Figure 8 polymers-14-04525-f008:**
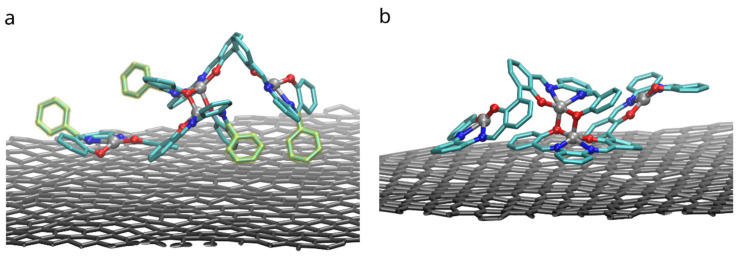
Configurations with the highest adhesion free energy for the dimers of (**a**) the compound A with functional phenyl groups and (**b**) compound B without them. Only upper graphene layer is shown and solvent molecules and compounds hydrogens are omitted for clarity. Phenyl functional groups are highlighted with yellow.

**Figure 9 polymers-14-04525-f009:**
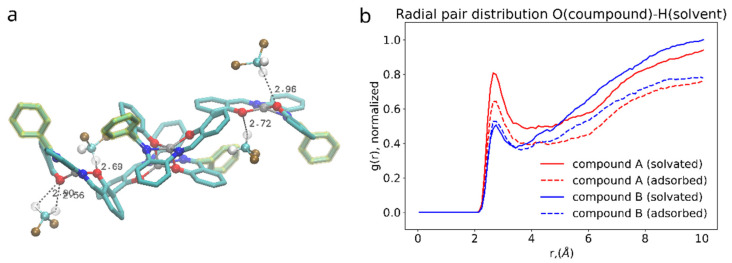
(**a**): 3D rendering of interaction between salphen’s oxygen and hydrogen of the DCM. (**b**): radial pair distribution function corresponding to this interaction for solvated (solid lines) and adsorbed (dashed lines) dimers.

**Figure 10 polymers-14-04525-f010:**
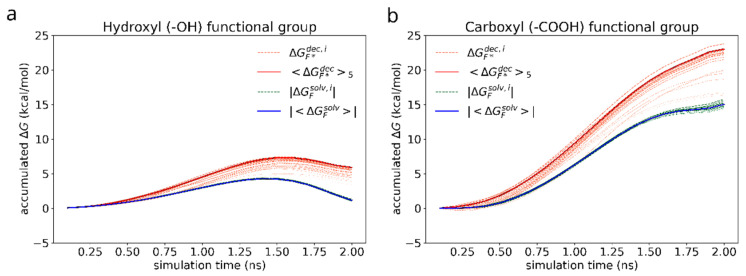
Accumulated free energy change during the decoupling (orange) and immersion (green) of the hydroxyl (**a**) and carboxyl (**b**) functional groups. The intensity of the color highlights how close the final free energy change during a particular simulation is to the maximum value over the whole set. Blue and red solid lines represent the average free energy change for the cases of solvated and adsorbed dimers (of N = 5 most favorable simulations) correspondingly. For the ease of visualization and comparison, absolute values are drawn for the solvation free energy of the functional groups.

**Figure 11 polymers-14-04525-f011:**
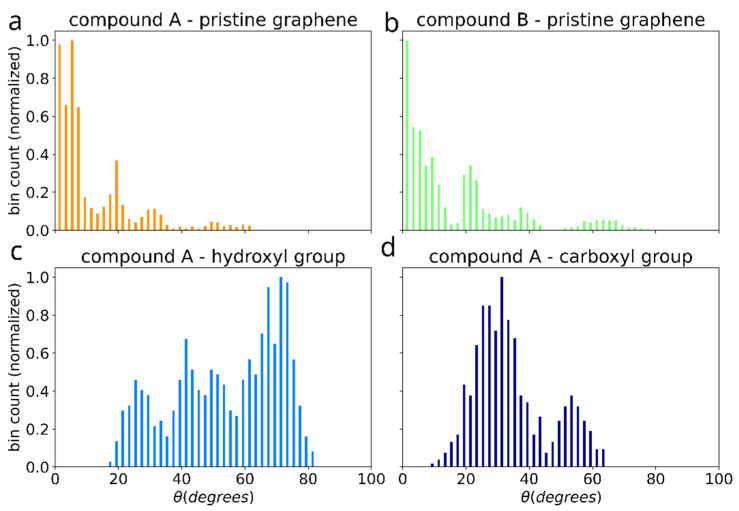
Distribution of the angle between the dimer axis (blue arrow in the Figure 3) and graphene plane for the cases of: pristine graphene with compounds A (**a**) and B (**b**), as well as for compound A adsorbed on graphene with -OH (**c**) and -COOH (**d**) functional groups.

**Figure 12 polymers-14-04525-f012:**
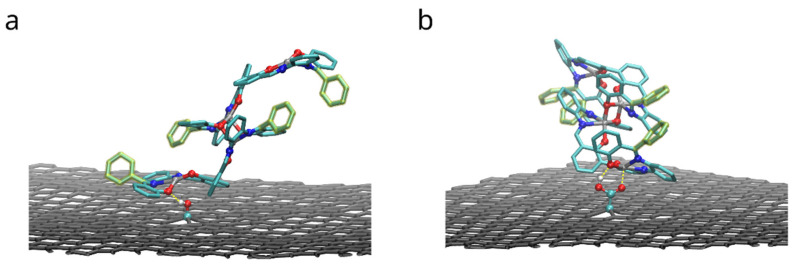
Configurations with the highest adhesion free energy for the dimers of the compound A with functional phenyl groups adsorbed on the surface of graphene functionalized with hydroxyl (**a**) and carboxyl groups (**b**). Only upper graphene layer is shown and solvent molecules and compounds hydrogens are omitted for clarity. Yellow dashed lines highlight hydrogen bonds and carboxyl oxygen interaction with zinc.

**Table 1 polymers-14-04525-t001:** Ab initio energy of adhesion (kcal/mol).

Functionalization Type	*E_adh_*	*E_adh_-E_adh_ (Pristine)*
Pristine	−38.8	-
B-doped	−40.7	−1.9
N-doped	−41.1	−2.2
=O	−35.5	+3.3
-OH	−40.5	−1.6
-COOH	−44.6	−5.6

**Table 2 polymers-14-04525-t002:** Average Gibbs free energies of solvation and adhesion of bis-(Zn)salphen dimers in DCM (kcal/mol). Confidence interval at 95% level for 5 independent simulations.

Compound:	Compound A	Compound B
Solvated (−$\Delta G_{S}^{dec}$)	−58.1 ± 5.1	−50.4 ± 1.2
Adsorbed (−$\Delta G_{S\&G}^{dec}$)	−67.7 ± 2.5	−63.7 ± 0.8
$\Delta G_{adh}^{}$	−9.7 ± 5.7	−13.7 ± 1.5

**Table 3 polymers-14-04525-t003:** Average Gibbs free energies of decoupling and immersion of functional groups on graphene surface with and without adsorbed compound dimer and its effect on adhesion of bis-(Zn) salphen dimers in DCM (kcal/mol). Confidence interval at 95% level for 5 independent simulations.

Functionalization:	Hydroxyl (-OH)	Carboxyl (-COOH)
Graphen & compound (−$\Delta G_{F*}^{dec}$)	−5.9 ± 0.1	−23.0 ± 1.1
$\mathrm{Graphene}\;(\Delta G_{F}^{solv}$)	−1.2 ± 0.1	−15.0 ± 1.0
$\Delta \Delta G_{adh}^{}$	−4.7 ± 0.2	−8.0 ± 1.4

## Data Availability

Data for the reproduction of the results reported in this study can be provided by the corresponding author, upon reasonable request.

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
