# Peer review of "Adhesion of Bis-Salphen-Based Coordination Polymers to Graphene: Insights from Free Energy Perturbation Study"

_polymers, 2022, doi:10.3390/polym14214525_

Round 1

Reviewer 1 Report

The manuscript "Adhesion of bis-salphen based coordination polymers to graphene: insights form free energy perturbation study" by Sergey Pyrlin, Veniero Lenzi, Luís Marques, Marta M.D. Ramos reports the computational (DFT) and modeling results of the interactions involved in the self-assembling of monomeric Zn-coordinated salphen, its dimer, the so-called bis-salphen and one more derivative (bis-salphen with an additional phenyl ring on each side of the dimer) on a graphene sheet. The latter has been considered as a monolayer, corresponding to the classical definition of graphene (a single layer of atoms arranged in a two-dimensional honeycomb lattice), as well as modeled as four layers, which could be referred to as graphite (although, the authors call it graphene too, erroneously to my point of view). The interaction energies, like H bonding, pi-pi stacking interactions, and energetics of adsorption are studied using FEP simulations. The authors come to the conclusion, that the negative values of the free energy changes coming from the interactions of the title compound with graphene surface also could also be in the same manner for the "sticky interactions" of such a compound with carbon nanotubes, and their stabilization in solutions, and the formation of ring-and-rod structures seen in experiments.

The idea of this simulational work is interesting, and the selected methods are appropriate. At the same time, there are serious flaws, which do not allow recommending this paper for publication in its present form.

1. In the abstract, the authors should mention which chemical modification has been studied. In fact, it is only one, therefore, should be precisely described.

2. Figure 1. Please either provide the chemical structure of the compound or draw the structure with a proper bond order.

3. The carbon nanotube surface has been modeled as a graphene monolayer or as four layers of graphite. In this regard, there are several questions.

a. was a monolayer continuous in DFT calculations due to periodic boundaries? how it can be then rectangular due to its symmetry?

b. was graphene (or four layers of graphene building graphite structure) periodic in MD? Was it frozen? If not, is this the reason, why 4 layers are modeled?

c. Which partial charges for 4 layers you have adopted for MD simulation if in DFT you have calculated only a monolayer). The same question is for monolayer and 4 layers, the upper layer of which has been with a chemical defect. If the graphene or 4 layers have been frozen, what about the integration of the equation of motion for these impurities?

4. Figures 2, especially 4, 5, 6, 7, and 8 have to be rearranged allowing the readers to understand what is written there. The font size is too small, and the resolution of the pictures is very low.

5. In Figure 3: do the authors understand the crystalline unit cell under a prism? If now, what is meant there?

6. Please mention on page 5, line 176, which solvent is modeled, you give this information later.

7. In the description of the model (in the main text or in the SI) please provide the cutoff radii for the interaction of the adsorbate molecules and the layers. We should understand how many graphite layers the molecule feels once adsorbed on the surface.

8. Tables 1 and 2 please use the same precision for all the numbers and the error bars. Or explain why you give different precision.

9. It would be interesting also to see the partial charges on bis-salphen after DFT, the "charged impurities" of graphene, as mentioned by authors in SI. Also, some authors provide the coordinates of the  DFT-relaxed structures in SI, it would be also desirable.

10. I see in this manuscript the problems with the analysis of the structural data. For example, please provide the lengths of the hydrogen bonding observed in the systems. Please measure the pi-pi stacking distance (this is especially needed since the statement is that (page 8, line 285) they "are mainly responsible..." The length of the stacking could be good proof that this is true. 

11. Figure 3. is this dimer bis-(Zn)salphen or dimer already functionalized with additional phenyl rings? Please clarify. 

12. Page 9, lines 308-309. Could it be, that these interactions are less favorable not because of the solvent, but mainly because of the non-planar structure of the compound with additional phenyl rings?

13. Figure 7. Please explain why the accumulated delta G decays after simulation time 1.50 ns, panel a. Why it is not the case on panel b. 

14. RDFs of the solvent around different parts of molecules A and B would be also helpful.

15. Do you think that the simulation time is enough? The question is why the molecules in Figure 8 are only a few atoms touching the surface. The authors also mentioned that the structures received in DFT are initial guesses for MD: Please provide more information, on how it was realized for the simulations shown in Figure 8.

16. Figure 8 - are these simulations on graphite or graphene? They are from MD trajectories, but suddenly only one (upper) layer is shown. It has to be corrected - in the figure caption or explained in the text.

I recommend reconsidering this manuscript after serious revision of the text and the figures.

Author Response

Dear Sir/Madam,

Thank you very much for your thorrough work to help us make the manuscript better.

Here are the responces for the points you've outlined:

  1. In the abstract, the authors should mention which chemical modification has been studied. In fact, it is only one, therefore, should be precisely described.

Implemented as suggested.

  1. Figure 1. Please either provide the chemical structure of the compound or draw the structure with a proper bond order.

Implemented as suggested.

  1. The carbon nanotube surface has been modeled as a graphene monolayer or as four layers of graphite. In this regard, there are several questions.
  2. was a monolayer continuous in DFT calculations due to periodic boundaries? how it can be then rectangular due to its symmetry?

Periodic boundary conditions were used in all calculations. A rectangular (not square) supercell of 6 by 10 rectangular unit cells of graphene hexagonal structure (initial dimensions approx. 24.8 x 25.8 Å) was used. An explanation of this is added to Methods section. A figure showing supercell in continuous graphene is added.

  1. was graphene (or four layers of graphene building graphite structure) periodic in MD? Was it frozen? If not, is this the reason, why 4 layers are modeled?

A periodic boundary conditions were used in all MD simulations as described in methods section. A rectangular supercell of 12 by 20 rectangular unit cells of graphene hexagonal structure (initial dimensions ~ 49.0 x 50.9 Å) was used. No atoms were frozen. 4 layers were selected as this way the adsorbed dimer only interacts with graphene atoms within cutoff of 10 Å for short range interactions and not with periodic image of solvent. An explanation of this is added to Methods section.

  1. Which partial charges for 4 layers you have adopted for MD simulation if in DFT you have calculated only a monolayer). The same question is for monolayer and 4 layers, the upper layer of which has been with a chemical defect. If the graphene or 4 layers have been frozen, what about the integration of the equation of motion for these impurities?

Zero partial charges were used for pristine graphene layers. For the modified layer the partial charges of the functional groups and their closest neighbors of the graphene carbon atoms were assigned to DFT derived charges. Integration of the equation of motion for all graphene (and functional group) atoms were performed using the Langevin thermostat as described in methods section. Partial charges used are added to SI.

  1. Figures 2, especially 4, 5, 6, 7, and 8 have to be rearranged allowing the readers to understand what is written there. The font size is too small, and the resolution of the pictures is very low.

Implemented as suggested.

  1. In Figure 3: do the authors understand the crystalline unit cell under a prism? If now, what is meant there?

The “prism” indeed originated from a unit cell of a 2D sheet of bis-(Zn)salphen structure as proposed in the previous article (Pyrlin et al. Soft Matter 2018). Here, however, it was used merely to describe the approximate shape of the dimer, to help “visualize” the ways the dimer initially placed on top of graphene surface. The fragment and figure 3 were revised to exclude the notion of “prism” and introduce the placement directly.

  1. Please mention on page 5, line 176, which solvent is modeled, you give this information later.

Reference to DCM as a solvent added to the beginning of the Methods subsection, describing MD simulations and to the description of initial model construction.

  1. In the description of the model (in the main text or in the SI) please provide the cutoff radii for the interaction of the adsorbate molecules and the layers. We should understand how many graphite layers the molecule feels once adsorbed on the surface.

Short range cut-off radius of 10 Å is added to the beginning of MD section.

  1. Tables 1 and 2 please use the same precision for all the numbers and the error bars. Or explain why you give different precision.

Implemented as suggested.

  1. It would be interesting also to see the partial charges on bis-salphen after DFT, the "charged impurities" of graphene, as mentioned by authors in SI. Also, some authors provide the coordinates of the  DFT-relaxed structures in SI, it would be also desirable.

Partial charges added to SI. In order not to overburden SI with DFT structures of salphen with graphene layers, the structures can be received from the corresponding author upon a reasonable request.

  1. I see in this manuscript the problems with the analysis of the structural data. For example, please provide the lengths of the hydrogen bonding observed in the systems. Please measure the pi-pi stacking distance (this is especially needed since the statement is that (page 8, line 285) they "are mainly responsible..." The length of the stacking could be good proof that this is true. 

Discussion of the DFT results in section 3.1 is extended to include separation distance and hydrogen bond lengths.

  1. Figure 3. is this dimer bis-(Zn)salphen or dimer already functionalized with additional phenyl rings? Please clarify.

More extended description added as suggested. Where relevant, the additional phenyl rings are highlighted with yellow.

  1. Page 9, lines 308-309. Could it be, that these interactions are less favorable not because of the solvent, but mainly because of the non-planar structure of the compound with additional phenyl rings?

Steric repulsion indeed contributes to weakening adhesion to graphene – comment added to the discussion. However, we also see from FEP calculations that phenyl functionalized compound also has a higher (by absolute value) solvation free energy. It can be linked to the fact that in the presence of these phenyl side groups there is an increased interaction between salphen’s O and H of DCM molecules. Hence the conclusion regarding the role of interaction with solvent. A discussion of this is added to the section 3.2.1

  1. Figure 7. Please explain why the accumulated delta G decays after simulation time 1.50 ns, panel a. Why it is not the case on panel b. 

Now Figure 10. A non-monotonous shape of the free energy change versus scaling parameter (lambda) results from the different rates at which Coulomb potential and dispersive and repulsive parts of soft Lennard-Jones potential scale with lambda, producing a net potential with a non-linear dependance on the parameter. Examples of non-monotonous dG vs t or lambda curve and analysis of separate contributions can be found in Klimovich 2015 (https://doi.org/10.1007/s10822-015-9840-9), Steinbrecher 2007 (https://doi.org/10.1063/1.2799191), Shirts 2005 (https://doi.org/10.1063/1.1873592).

In case of -COOH much stronger electrostatic attraction to solvent molecules completely dominates the behavior of the total curve. A brief comment on this matter was added to the text. A figure S1 in the supplementary information shows side by side another FEP simulation of immersion of the -OH (left) and -COOH (right) groups, where scaling of the Lennard-Jones and Coulomb potentials was performed sequentially (red curves). It is clear that while the shape the dG curve for such simulation is different from the older simulation (blue curve), the net change of the free energy stays the same.

  1. RDFs of the solvent around different parts of molecules A and B would be also helpful.

Added to SI as figure S2.

  1. Do you think that the simulation time is enough? The question is why the molecules in Figure 8 are only a few atoms touching the surface. The authors also mentioned that the structures received in DFT are initial guesses for MD: Please provide more information, on how it was realized for the simulations shown in Figure 8.

The dimers shown in the figure (now replaced with figures 3, 8, 11 and 12) illustrates the possible orientations of the dimer on graphene surface: the dimer “lying” flat on graphene or “standing” interacting with graphene more closely with one side with aromatic rings aligned parallel to the surface, while the rest of the molecule is surrounded by solvent (not shown in the figure). The distribution of dimer orientations serve to illustrate that on flat graphene surface bis-salphen dimers are more likely to adopt the “lying” orientation even though initially both orientations were present equally. On the contrary, interaction with -OH or -COOH functional group stabilizes the “standing” orientation. A better discussion of this aspect is added to the text.

  1. Figure 8 - are these simulations on graphite or graphene? They are from MD trajectories, but suddenly only one (upper) layer is shown. It has to be corrected - in the figure caption or explained in the text.

Same 4-layered graphite structure was used in all MD simulations. Only one layer is shown for visual clarity. Explanation of this is added to the figures’ caption.

Reviewer 2 Report

Authors present modelling results on the polymer adhesion to graphene including the free energy perturbation method and all-atom molecular dynamics simulations. The work is interesting and fits the general scope of Polymers. However, the present manuscript requires modifications before being in publishable form. These are mainly on how the methodology and results are presented.

For better reading flow authors could move all the (technical information) contained in S.I. into the main text. Currently the very limited size of the S.I. along with the balanced one of the main manuscript allows for this.

In some instances the meaning of sentences is unclear and it has to be improved. For example: lines 173-175: “For each … relaxed”. What is meant by “dimer was initiated” repeated numerous times in the description of the method? Do the authors mean “was placed” ? The manuscript requires a careful check in grammar and syntax.

Authors use a different approach for the functionalized graphene avoiding the slow decoupling process. However, it is not clear which is the corresponding discrepancy of the two approaches. Have the authors performed any simulation incorporating the decoupling to compare the results?

Why the authors have used the OPLS-3 force field for the description of interactions for the all-atom MD simulations, instead of more recent ones? Authors should provide a small explanation on this. Is it based on the Schiff base parametrization as written in the text?  

This is also relevant as the authors compare the present results with older ones obtained through a different interaction potential.

Figure 2 could be presented in higher quality as the different variations of the compound are very difficult to be distinguished visually. Authors could zoom to the part of the (Zn)salphen-based compound and the functionalization of the sheet.

Figure 4: The description in the legend is rather telegraphic. The individual panels (a) to (d) could be explained in more detail. Same for Figure 5 which corresponds to the functionalized graphene.

In Figure 6, font size is too small to read the physical variables shown. There is also confusion with the line “Correspondingly, … favorable simulations”. The sentence is incomplete (black and red corresponds to which different sets?) also because the legend is not sufficiently clear due to the font size. Colors (especially blue against black) could be chosen different to have better contrast. The small font size problem also persists in all successive figures.

Line 93: “a very precise results” -> “very precise results”.

Line 140: “functionelized” -> “functionalized”.

Line 185: “were gradually” -> “was gradually”.

Line 221: “decoupling the” -> “decoupling of the”.

Line 249: “are mostly” -> “is mostly”.

Line 298: “energy vary” -> “energy varies”.

Line 268: “with impact” -> “with the impact”.

Comma is wrongly used after the word “fact” (i.e. as in lines 304 and 308).

Line 311: “decrease” -> “decreases”.

Line 347: “to aligned” -> “to be aligned”.

Line 355: Sentence is unclear.

Author Response

Dear Sir or Madam,

Thank you very much for you thorrough work to help us make the manuscript better.

Here are the responces to the points you've outlined:

  1. For better reading flow authors could move all the (technical information) contained in S.I. into the main text. Currently the very limited size of the S.I. along with the balanced one of the main manuscript allows for this.

Technical information was expanded and moved to main text

  1. In some instances the meaning of sentences is unclear and it has to be improved. For example: lines 173-175: “For each … relaxed”. What is meant by “dimer was initiated” repeated numerous times in the description of the method? Do the authors mean “was placed” ? The manuscript requires a careful check in grammar and syntax.

Indeed “was placed” was in mind. Replaced as suggested.

  1. Authors use a different approach for the functionalized graphene avoiding the slow decoupling process. However, it is not clear which is the corresponding discrepancy of the two approaches. Have the authors performed any simulation incorporating the decoupling to compare the results?

One of the reasons behind switching the approach, is that the former approach (decoupling the whole bis-salphen dimer from the whole surrounding) is less stable in case of functional group: Zn-salphen adhesion to -OH and -COOH is mostly due to electrostatic interaction, which is turned off first. As the result, without introduction of virtual constraints the dimer would break-off the functional group at a random moment in the beginning of a FEP simulation causing large discrepancy in FEP results between independent runs. Decoupling the functional group while keeping the dimer attracted to the graphene surface provides additional stability. Therefore, the direct validation of one approach by another is not a reliable option. An alternative way would be to compare both approaches to a well converged PLUMED free energy scan, this however requires a separate high HPC load simulation a large part of which is unnecessary. We believe that selecting a tailored approach for each task is sufficient to support the conclusions we make. A brief discussion of this issue is added to the section 3.2.1.

  1. Why the authors have used the OPLS-3 force field for the description of interactions for the all-atom MD simulations, instead of more recent ones? Authors should provide a small explanation on this. Is it based on the Schiff base parametrization as written in the text?  This is also relevant as the authors compare the present results with older ones obtained through a different interaction potential.

In fact, the initial reference to OPLS-3 was mistake. Most of the parameters used correspond to the core OPLS-AA. The exception was parameters describing the Schiff base framgent (C-N=C-C), which were selected from OPLS_2005 as giving closer agreement to DFT results. While this is even older parametrization, it is still used in many recent papers. A brief section regarding this is added to the text.

  1. Figure 2 could be presented in higher quality as the different variations of the compound are very difficult to be distinguished visually. Authors could zoom to the part of the (Zn)salphen-based compound and the functionalization of the sheet.

Figure moved to discussion section (now Figure 6) and replaced as suggested.

  1. Figure 4: The description in the legend is rather telegraphic. The individual panels (a) to (d) could be explained in more detail. Same for Figure 5 which corresponds to the functionalized graphene.

More extended description is added to Figures 4 and 5.

  1. In Figure 6, font size is too small to read the physical variables shown. There is also confusion with the line “Correspondingly, … favorable simulations”. The sentence is incomplete (black and red corresponds to which different sets?) also because the legend is not sufficiently clear due to the font size. Colors (especially blue against black) could be chosen different to have better contrast. The small font size problem also persists in all successive figures.

Now figure 7. Font size increased. Description improved. Colors changed.

  1. Grammar & style corrections:
    1. Line 93: “a very precise results” -> “very precise results”.
    2. Line 140: “functionelized” -> “functionalized”.
    3. Line 185: “were gradually” -> “was gradually”.
    4. Line 221: “decoupling the” -> “decoupling of the”.
    5. Line 249: “are mostly” -> “is mostly”.
    6. Line 298: “energy vary” -> “energy varies”.
    7. Line 268: “with impact” -> “with the impact”.
    8. Comma is wrongly used after the word “fact” (i.e. as in lines 304 and 308).
    9. Line 311: “decrease” -> “decreases”.
    10. Line 347: “to aligned” -> “to be aligned”.
    11. Line 355: Sentence is unclear.

Implemented as suggested.

Round 2

Reviewer 1 Report

I do not have further questions and recommend this manuscript for publication

Author Response

Dear Reviewer,

Thank you very much for helping us make the manuscript better.

Sergey Pyrlin

Reviewer 2 Report

Authors have modified their manuscript according to the feedback received with respect to the original version. The manuscript has been significantly improved and is thus in publishable form for the journal “Polymers”.

I have only some remaining comments:

) Title: “form free energy” –> “from free energy” (this is important).

) Line 16: “tell about” -> “reveal about”.

) Line 70: “was left without investigation” -> “remains unexplored”.

) Line 109: “than be” -> “then be”.

) Line 209: “good description” -> “accurate description”.

) Line 253: “were first simulated using in” -> “was simulated in”; “to then” -> “, then”.

) Line 425: “distorts otherwise” -> “distorts the otherwise”.

Author Response

Dear Reviewer,

Thank you very much for helping us make the manuscript better.

Your corrections implemented as suggested:

) Title: “form free energy” –> “from free energy”.

) Line 16: “tell about” -> “reveal about”.

) Line 70: “was left without investigation” -> “remains unexplored”.

) Line 109: “than be” -> “then be”.

) Line 209: “good description” -> “accurate description”.

) Line 253: “were first simulated using in” -> “was simulated in”; “to then” -> “, then”.

) Line 425: “distorts otherwise” -> “distorts the otherwise”.

Sergey Pyrlin